# Nutritional Regimes Enriched with Antioxidants as an Efficient Adjuvant for IBD Patients under Infliximab Administration, a Pilot Study

**DOI:** 10.3390/antiox11010138

**Published:** 2022-01-08

**Authors:** Marina Liso, Annamaria Sila, Giulio Verna, Aurelia Scarano, Rossella Donghia, Fabio Castellana, Elisabetta Cavalcanti, Pasqua Letizia Pesole, Eduardo Maria Sommella, Antonio Lippolis, Raffaele Armentano, Anna Maria Giudetti, Daniele Vergara, Pietro Campiglia, Rodolfo Sardone, Margherita Curlo, Mauro Mastronardi, Katia Petroni, Chiara Tonelli, Angelo Santino, Marcello Chieppa

**Affiliations:** 1National Institute of Gastroenterology “S. de Bellis”, Institute of Research, 70013 Castellana Grotte, Italy; marina.liso@irccsdebellis.it (M.L.); a.sila@outlook.it (A.S.); rossella.donghia@irccsdebellis.it (R.D.); fabio.castellana@irccsdebellis.it (F.C.); elisabetta.cavalcanti@irccsdebellis.it (E.C.); pesoleletizia@gmail.com (P.L.P.); antonio.lippolis@irccsdebellis.it (A.L.); raffaele.armentano@irccsdebellis.it (R.A.); rodolfo.sardone@irccsdebellis.it (R.S.); margherita.curlo@irccsdebellis.it (M.C.); mauro.mastronardi@irccsdebellis.it (M.M.); 2Department of Pharmacy, School of Pharmacy, University of Salerno, 84084 Fisciano, Italy; gverna@unisa.it (G.V.); esommella@unisa.it (E.M.S.); pcampiglia@unisa.it (P.C.); 3Department of Medicine, Digestive Health Research Institute, Case Western Reserve University School of Medicine, Cleveland, OH 44106, USA; 4Unit of Lecce, Institute of Sciences of Food Production C.N.R., 73100 Lecce, Italy; aurelia.scarano@ispa.cnr.it; 5Department of Mathematics and Physics “Ennio De Giorgi”, University of Salento, via Monteroni, 73100 Lecce, Italy; anna.giudetti@unisalento.it (A.M.G.); daniele.vergara@unisalento.it (D.V.); 6Department of Biosciences, University of Milan, via Celoria 26, 20133 Milan, Italy; katia.petroni@unimi.it (K.P.); chiara.tonelli@unimi.it (C.T.); 7Dietetics and Clinical Nutrition Laboratory, Department of Public Health, Experimental and Forensic Medicine, University of Pavia, 27100 Pavia, Italy

**Keywords:** antioxidant, anthocyanins, IBD, diet, infliximab, pilot study

## Abstract

Antioxidants are privileged candidates for the development of adjuvants able to improve the efficiency of pharmacological therapies, particularly for chronic inflammatory syndromes. During the last 20 years, anti-TNFα (tumor necrosis factor alpha) monoclonal antibodies infusion has been the biological therapy most frequently administered but there is still large space for improvement in disease remission rates and maintenance. In this context, nutritional bioactive compounds contained in dietary patterns or included as supplements, may act as adjuvants for the induction and maintenance of IBD (inflammatory bowel diseases) remission. To verify this possibility, a single-center preliminary study (SI-CURA, Soluzioni Innovative per la gestione del paziente e il follow up terapeutico della Colite UlceRosA) was designed and carried out to evaluate whether a daily administration of purple corn supplement could improve the response to Infliximab (IFX) infusion of IBD patients with both Crohn’s disease (CD) and ulcerative colitis (UC). A cohort of 47 patients was enrolled in the study. Biological samples were collected before the first and the third IFX infusion. All patients received nutritional guidelines, 27 of them received commercial red fruit tea with low anthocyanins content, while 20 received a purple corn supplement with a high anthocyanin content. Results show that the administration of an antioxidant-enriched purple corn supplement could improve IFX-mediated disease remission in terms of circulating inflammatory markers. Comparison between CD and UC patients revealed that, at this anthocyanin dosage, the purple corn extract administration improved the IFX response in CD but not in UC patients. Our results may pave the way for a new metacentric study of CD patients, recruiting a wider cohort and followed-up over a longer observational time.

## 1. Introduction

Inflammatory bowel diseases (IBD), including Crohn’s Disease (CD) and Ulcerative Colitis (UC), are multifactorial disorders characterized by chronic inflammation of the gut, abdominal pain, diarrhea, and rectal bleeding. Inflammation in CD involves the entire gastrointestinal tract, whereas in UC only the colon and the rectum are affected. Transmural inflammation is typical of CD, while in UC the inflammation is confined to the mucosa [1,2,3].

During the past few decades, IBD treatment has made remarkable advances, mainly related to the availability of monoclonal antibodies to block several elements of the immune and inflammatory response. IBD patients were originally treated with corticosteroids and thiopurines, and basically no other therapeutic options were available until the introduction of the first hybrid anti-tumor necrosis factor alpha (TNFα)-antibody in 1995 [4].

Currently, the administration of infliximab (IFX), a chimeric monoclonal antibody against TNFα, is a common strategy even used as the first or initial therapy, as it can induce mucosal healing and reduce the need for surgical intervention more frequently than corticosteroids or thiopurines [5,6]. Despite these results, IFX administration is frequently unable to induce long-term IBD remission, as a significant percentage of patients fail to respond and others experience a secondary loss of response [7]. As most patients are young at IBD onset, treatment goals are typically focused first on inducing disease remission and then on maintaining clinical remission and preventing the onset of complications.

Epidemiological data clearly indicate that the IBD prevalence is increasing in Eastern societies, in parallel with the change in lifestyle habits and diet. Nutritional components of the diet can favor the growth of specific bacterial strains, triggering disease progression [8,9,10,11].

Western diets often have a poor content of anthocyanins and polyphenols, natural anti-inflammatory molecules that can potentially be used as adjuvants to the anti-TNFα therapy in the form of extracts from functional foods. Epidemiological studies demonstrated that the consumption of anthocyanins-rich foods can protect from numerous diseases including cancer, cardiovascular disease, and age-related neurodegenerative diseases, all characterized by deregulated inflammation status [12,13,14,15,16,17].

In particular, it is well known that polyphenols suppress inflammatory cytokine secretion by immune cells exposed to bacterial components [18,19,20]. Polyphenols, such as quercetin, are potent iron sequestering agents that can reduce iron concentration by inducing the expression of Slpi (Secretory leukocyte protease inhibitor) and HO-1 (Heme oxygenase-1) [21,22,23,24]. These proteins, in turn, can block the inflammatory cascade and free radicals formation [25]. Furthermore, the administration of a polyphenol-enriched diet was able to modify the intestinal microbiota diversity in a murine model of IBD [26,27,28], thus suggesting that specific nutritional regimes may block the trigger for disease recurrence in IBD patients.

Oxidative stress is a well-accepted component of the IBD pathology, often considered a result of the ongoing chronic inflammation, even though recent data demonstrated a lower antioxidant capacity of patients in remission versus the control group [29]. Furthermore, recent data suggest that diet and an abundance of microbial species may boost the generation of pro-oxidant molecules in the gastrointestinal tract [30].

In light of these results, we designed a pilot interventional study to address the possibility of using antioxidant-enriched complementary approaches to improve IBD patients’ clinical response to IFX infusion, and reduce the risks of secondary loss of response. In the study group, we administered a daily single dose of anthocyanin-rich supplement obtained from purple corn. This supplement is a high-anthocyanin water-soluble extract, obtained from *B1 Pl1* purple corn cobs. In previous studies, the purple corn supplement was reported to protect from myocardial damage induced by Doxorubicin, an anthracycline widely used as a chemotherapeutic drug against a variety of cancer types [31]. Administration of purple corn extract reduced the development of trigeminal inflammatory pain in rats and decreased obesity-associated inflammation, by preventing recruitment and proliferation of adipose tissue macrophages and promoting their shift towards an anti-inflammatory phenotype [32,33].

## 2. Materials and Methods

### 2.1. Study Design

This single-center study was based on the administration of a purple corn supplement to IBD patients (Crohn’s disease and ulcerative colitis) receiving infliximab between September 2019 and December 2020 at the Division of Gastroenterology and Digestive Endoscopy of the National Institute of Gastroenterology “S. de Bellis”, Castellana Grotte, Bari, Italy.

The experimental design of our study consisted of nutritional intervention in a heterogeneous group of 47 IBD patients (22 with CD and 25 with UC) at the start of the IFX infusion (Figure 1). The day of the first infusion is denominated time zero (T0) hereafter in the study. Patients were randomly subdivided into 2 groups: the diet + red fruit tea (RFT) or diet + purple corn extract (RED). The first group of patients received a normocaloric nutritional regime and a daily dose of commercial red fruit tea (total polyphenols content about 2 mg GAE/g DW (gallic acid equivalents per g of dry weight); total anthocyanin content of 0.5 mg cyanidin 3-glucoside (C3G) equivalents/g DW), the second group received the same normocaloric diet plus a daily dose of purple corn extract (with a flavor similar to red fruit tea) [31,33]. The RFT group included 27 patients (15 males and 12 females), 13 affected by CD and 14 by UC. The RED group included 20 patients (8 males and 12 females), 9 with CD and 11 with UC. Patients’ disease was monitored before each IFX infusion, and blood samples were collected before the first and the third anti-TNF infusion (T0 and T3, respectively), as shown in Figure 1.

A comprehensive assessment of patients’ circulating biomarkers and abundance of inflammatory mediators was performed before the T0 and T3, together with routine disease assessments.

The treatment goal, after 8 weeks of therapy (T3), was a decrease by at least 6 points in the Mayo and Montreal scores [34,35], independently of the starting score. Inclusion criteria were: patients scheduled treatment with anti-TNFα therapy, therapy to be performed with either Remicade^®^ (Janssen, Beerse, Belgium) or the biosimilar Remsima^®^ (Celltrion, Incheon, Korea), (5 mg/kg body weight), and no previous treatment with anti-TNFα therapy. Exclusion criteria were comorbidities with other diseases (assessed with the Charlson Comorbidity Index) [36], pregnancy or breastfeeding, a malignant neoplasm in the last 10 years, ongoing immunosuppressive or immunomodulatory therapy, and the need for artificial nutrition.

### 2.2. Ethics Statement

The investigation was conducted following the ethical standards, the Declaration of Helsinki, and international guidelines, and had been approved by the authors’ institutional review board (Number 333, 31 July 2019). All patients provided written informed consent to take part.

### 2.3. Assessment of Participants

Study subjects’ initial evaluation included a medical history, physical examination, and lab tests, including complete blood count (CBC) with differential C-reactive protein (CRP), albumin, iron, ceruloplasmin, alpha-1 antitrypsin, transferrin, vitamin D, ferritin, folates, and vitamin B12.

Patients underwent clinical follow-up at 2 and 8 weeks. Standardized questionnaires about the general symptoms and adverse effects were completed during each study visit [25]. In addition, patients underwent a physical examination, and the same blood work was performed at the enrollment and follow-up visits.

### 2.4. Dietary Therapy Management

A total of two dietary regimes were adopted. The first group followed a normocaloric diet consisting of a maximum daily intake of 2075.70 Kcal, supplemented with a daily intake of one cup of a commercial red fruit tea (RFT); the other group followed the same normocaloric diet supplemented with a daily cup of purple corn extract (RED). The diet consisted of about 18.81% of proteins, 22.34% of fats, 58.54% carbohydrates [37,38], and was created using the software WinFood 3.29 software (Medimatica Srl Unipersonale, Colonnella, Italy). The detailed bromatological composition is indicated in Appendix A.

At the enrollment following the nutritional care process, the nutritional assessment was performed with the recall-24h diary. Then, patients’ anthropometric parameters (weight, height, Body Mass Index, and waist, calf, and triceps circumference) and blood pressure were recorded. During the first visit, patients filled out the MNA (Mini Nutritional Assessment) and MUST (Malnutrition Universal Screening Tool) questionnaires [39,40]. Patients were randomly assigned to the RFT or the RED group and started the nutritional regime. When enrolled in the study, all patients were instructed how to correctly prepare the supplement breakfast infusion, thus the nutritionist was not blinded, whereas the physicians and final analysts were blinded. The RFT or RED supplements were taken every day at breakfast. Patients were strongly advised to follow the nutritionist’s suggestions, to introduce new foods gradually and in a sequential manner, according to their symptoms and necessities. The nutritional regime consists of a balanced Mediterranean Diet with a total fibers content based on the Recommended Dietary Allowance (RDA) and a great variety of foods; seasonal fruit and vegetables were strongly recommended. The goal of the nutritional intervention was to educate patients about a healthy and balanced diet, to treat any eventual ongoing malnutrition, and to prevent IBD disease recurrence and consequent hospitalization.

### 2.5. Supplement Composition

The supplement is extracted from *B1 Pl1* purple corn cobs using ethanol:H_2_O (30:70 *v/v*) at 55 °C for 1 h, titrated to a concentration of 4% anthocyanins and spray-dried to obtain water-soluble granules packaged in single-dose packs (Sveba Srl, Appiano Gentile, CO, Italy) [31,33]. The RED was drunk once a day, offering a 125 mg total daily dose of anthocyanins per day, already proven to be safe and well-tolerated [41].

### 2.6. Sample Collection and Cytokine Detection

Before and during the follow-up visit, 10 mL of blood were collected and stored at −80 °C in the institute biobank. Circulating cytokine detection was performed using the ELISA kit (R&D Systems, Minneapolis, MN, USA) for IL-1β and “MACSPlex Cytokine 12 Kit human” (Miltenyi Biotec, Bergisch Gladbach, Germany) for GM-CSF, IFN-α, IFN-γ, IL-2, IL-4, IL-5, IL-6, IL-9, IL-10, IL-12p70, IL-17A, and TNFα, following the manufacturer’s instructions. Flow Cytometry acquisition was performed using NAVIOS (Beckman Coulter, Brea, CA, USA). Flow cytometry analysis was performed using Kaluza Software 1.5 (Beckman Coulter, Brea, CA, USA).

### 2.7. Statistical Analysis

The entire sample consisted of 47 subjects assessed at baseline (T0) and at follow-up time (T3, after 8 weeks of therapy). All data are shown as mean ± SD, median (min to max) for categorical variables, and as N (%) for categorical ones. The normality of distributions was assessed with the Shapiro–Wilk test. A non-parametric paired approach was chosen to test differences between the T0 and T3 observations (Wilcoxon sum rank test for paired samples). A *p*-value ≤ 0.05 was considered statistically significant. Mann–Whitney sum rank test and Fisher’s exact test were performed to assess statistically significant differences between the RFT and RED groups for continuous and categorical variables, respectively.

## 3. Results

### 3.1. Baseline Characteristics

Clinical and pathological features of the enrolled patients are depicted in Table 1. There was a high grade of heterogeneity in terms of age and gender in our sample cohort. We enrolled 22 CD patients and 25 UC patients; 42.55% were smokers.

The disease phenotype according to the Montreal Classification was inflammatory in 54.55% of our CD patients, stricturing in 27.27%, and penetrating in 18.18%. A total of 13.64% of CD patients had ileal disease while 4.55% had a colic phenotype and the majority of them (81.81%) had an ileocolic phenotype. All the UC patients (100%) had a pancolitis disease phenotype.

Before the enrollment, 46.81% of the patients had been taking ongoing steroid therapy.

### 3.2. Inflammatory Biomarkers Modulation after 8 Weeks of Therapy

Patients were enrolled in the study on the day of the first IFX infusion (T0). All 47 patients completed the study and no dropout occurred.

Appendix A and Table 2 show patients’ characteristics and serological data at T0 and after 8 weeks of therapy (T3).

None of the 47 patients recruited was a primary non-responder, so the entire group was able to complete the experimental protocol until the T3 (third infusion).

Unsurprisingly, IFX administration resulted in a reduction of inflammatory scores (both significant, CDAI, and Mayo) and serological markers, independently of which antioxidant-rich supplement was administered. Importantly, T3 patients also displayed increased concentration of iron and folate (Appendix A), most likely due to the reduced intestinal permeability, blood loss, and watery stools as a consequence of the IFX-mediated mucosal healing [42]. Circulating TNFα concentration was reduced in patients treated with IFX, when considering the mean of TNFα levels, although the concentration in several patients was under the detection limit even before the onset of IFX administration.

Serum analysis at T3 revealed differences between the RFT and RED groups of patients. Appendix A, Table 3 and Appendix A show an analysis of the diet with commercial red fruit tea (RFT) vs. the diet with purple corn extract (RED) group at T3. Despite IFX being administered to both groups, CRP, ceruloplasmin, and TNFα were significantly more reduced in the group receiving the RED. The increased folate circulating levels, following IFX administration, were not significantly different between the groups at T3 (*p* = 0.06), although an evident increase was observed in the RED group. Conversely, the iron concentration decreased, but not significantly, in the RED group as compared to the RFT group at T3.

Diet may act differently on CD or UC. Figure 2, Table 4 and Appendix A show the impact of RFT and RED on CD, while Appendix A, Table 5 and Appendix A show the results in UC patients.

Despite the reduced number of patients in the groups, results indicated that in CD patients several inflammatory cytokines were significantly reduced, including GM-CSF (*p* = 0.04), IFN-γ (*p* = 0.02), TNFα (*p* = 0.04), IL-5 (*p* = 0.04), IL-9 (*p* = 0.04), IL-10 (*p* = 0.02), IL-12p70 (*p* = 0.02), and IL-17A (*p* = 0.01). Together with these results, the inflammatory biomarker CRP was also significantly reduced by the RED supplement (*p* = 0.03) while iron circulating concentrations were unchanged and ceruloplasmin decreased, but not significantly (*p* = 0.06). Finally, the observed folate increase was non-significant, likely due to the elevated SD in this small sub-group of patients. In UC patients, the only significant variation was a reduction in the BMI, while no inflammatory biomarker or circulating cytokine resulted significantly affected by RED.

## 4. Discussion

The introduction of biological drugs has been a pivotal moment for IBD treatment, and in particular, the anti-TNFα infusion has become a standard strategy in clinical practice. Disease remission, mucosa healing, and clinical improvement are standard goals for patients receiving IFX infusion. Nonetheless, numerous patients fail to respond to IFX or become secondary non-responders over time.

Currently, the vast majority of research effort is devoted to the development of new drugs, while marginal attention has been paid to adjuvant strategies, particularly nutritional-derived intervention. The gastrointestinal (GI) tract is a complex microenvironment; the molecular crosstalk among the human host, luminal resident microbes, and nutritional intakes plays a pivotal role in physiology, in both physiological and pathological conditions. Epidemiological data indicate the role of diet in IBD onset and relapse, but dietary adjuvant intervention to induce or maintain IBD remission has been little considered in many clinical studies [43,44,45].

The GI tract surface area is by far the largest border of the body [46] underlined by immune cells, particularly macrophages and dendritic cells (DCs), to patrol the epithelial cell barrier [47]. DCs under the columnar villus epithelium extend processes across this epithelium and interact with the intestinal lumen content, bacteria, food-derived products, and metabolites [48]. During the last decade, we have investigated several aspects of the crosstalk between immune cells and nutritional-derived bioactive compounds and microelements [18,19,20,49]. With the aim of identifying nutritional-derived bioactive components able to act as adjuvants in IBD treatments, our research groups explored the effects of a large set of antioxidant compounds belonging to the large polyphenols family using both in vitro and in vivo models of gut inflammation [21,22,23,24,25,26,27,28,50,51].

Commercial red fruit tea or purple corn supplement, obtained from *B1 Pl1* purple corn cobs, were randomly assigned to IBD patients, both UC and CD, starting on the day of the first IFX infusion. Even in light of the limited number of patients recruited, these data are coherent with previous studies demonstrating an inflammatory suppression effect induced, in vitro and in vivo, by RED administration [31,32,33].

Our IBD patients cohort was heterogeneous, including UC and CD patients. The present preliminary study aimed to evaluate whether the adjuvant intervention could provide benefits to the vast majority of IBD patients and identify the best responders to this specific antioxidant-enriched strategy. Our results indicated that a daily intake of about 125 mg of purple corn anthocyanins could be effective in suppressing intestinal inflammation and consequently reducing the intestinal leaky barrier characterizing IBD patients [52].

The supplement administration had a different impact in CD or UC patients. Indeed, the analysis of serological biomarkers at the follow-up observation indicates that the impact of the RED extract was mostly relevant in the CD group, where several inflammatory cytokines and CRP were found to be significantly reduced. Our results may indicate that the biological activity of purple corn anthocyanins is exerted mainly in the small intestine. Immune cells exposed to anthocyanins lose the ability to produce and release inflammatory mediators. Immune cells protrusions across the epithelial barrier are more frequent in the ileum as compared to the colon [53,54]. Furthermore, inflammatory conditions increase the rate of DCs protrusions from the small intestinal villi [46]. Thus, it is possible that, in CD more than in UC patients, intestinal resident DCs could be exposed to purple corn anthocyanins. Another possibility is that the intestinal microbiota may metabolize purple corn anthocyanins, progressively reducing the concentration of bioactive compounds, or else water absorption may reduce their bioavailability, particularly in the distal colon. Therefore, UC patients may require a higher dosage of adjuvant administration or different formulation of the adjuvant.

Overall, the results of the present preliminary study encourage a new larger study based on the administration of purple corn cobs extracts to CD patients starting IFX therapy. Together with the chronic inflammatory syndromes, IBD patients run the risk of developing micronutrient deficiencies. It is important to underline that our results indicate that patients following balanced diets and antioxidant administration show improved iron and folate levels, together with a reduction of numerous inflammatory biomarkers.

## 5. Conclusions

The present study demonstrated that the purple corn extract (RED) was able to improve patients’ responses to IFX administration, as demonstrated by the reduction of the CRP and ceruloplasmin in the purple corn supplemented group. Our data also make it clear that this supplement was able to reduce inflammatory biomarkers in CD but not in UC patients. Our results may pave the way to a wider multicentric clinical trial focused on supplements administration to CD patients as an adjuvant to anti-TNFα therapy, to improve the disease remission rates and reduce the risk of disease relapse during the maintenance therapy.

Currently, there is some confusion about what dietary intervention may improve the induction and preservation of remission in IBD and their use is challenging. Our single-center study provides the rationale for the design of a specific adjuvant intervention for CDs patients starting the IFX therapy. Further studies will clarify whether different strategies or increased antioxidant concentration may also improve the efficiency of the same supplement in UC patients.

## Figures and Tables

**Figure 1 antioxidants-11-00138-f001:**
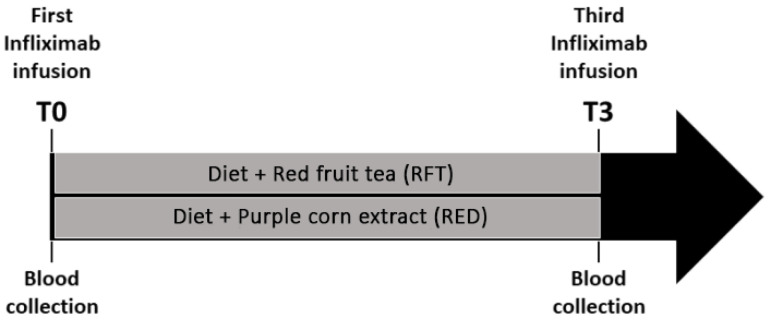
Experimental design: 47 enrolled IBD patients, before the first IFX infusion, were randomly assigned to 2 treatment groups (diet + RFT or RED).

**Figure 2 antioxidants-11-00138-f002:**
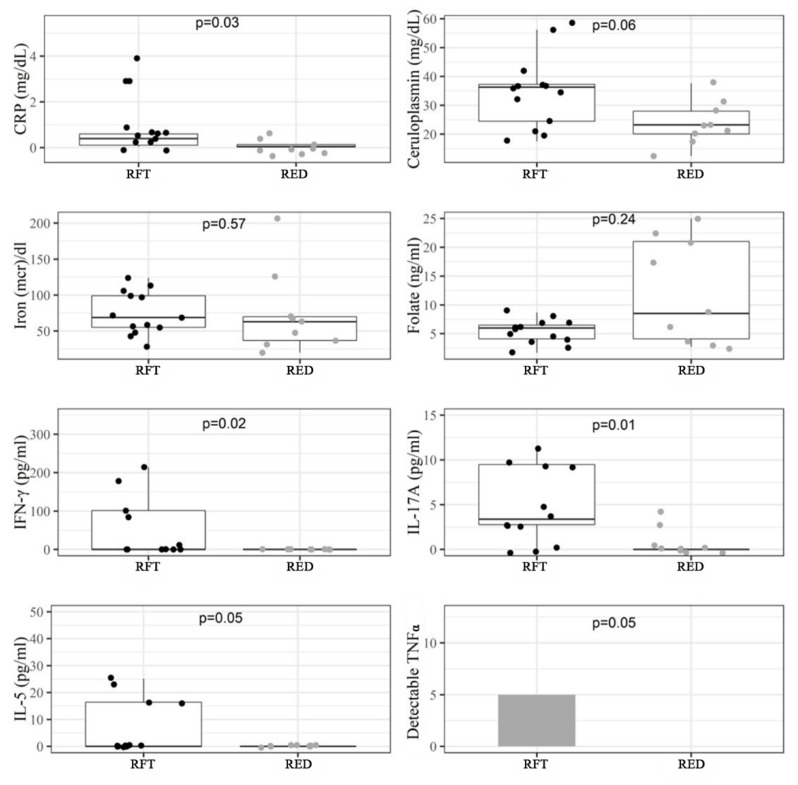
Serological biomarkers that were different between the red fruit tea (RFT) and purple corn extract (RED) groups in CD patients at T3 (N: 22).

**Table 1 antioxidants-11-00138-t001:** Clinicopathologic features of enrolled patients (N: 47).

	T0	
	Mean ± Sd	Median (Min to Max)
Proportions N (%)	47 (100.00)	
Age (years)	42.12 ± 14.44	42.5 (17 to 70)
Gender N (%)		
Male	24 (51.10)	
Female	23 (48.90)	
Disease N (%)		
CD	22 (46.80)	
UC	25 (53.20)	
Smokers N (%)	20 (42.55)	
Time from diagnosis to starting IFX (months)	78.19 (58.18)	64 (10 to 240)
Type of Disease N (%)-Montreal Classification		
B1-Inflammatory	12 (54.55)	
B2-Stricturing	6 (27.27)	
B3-Penetrating	4 (18.18)	
Location of CD disease patients N (%)-Montreal Classification		
L1 Ileal	3 (13.64)	
L2 Colic	1 (4.55)	
L3 Ileocolic	18 (81.81)	
Location of UC disease patients N (%)-Montreal Classification		
E1 Ulcerative	0 (0.00)	
E2 Left sided UC (distal UC)	0 (0.00)	
E3 Extensive UC (pancolitis)	25 (100.00)	
Steroid therapy (%)	22 (46.81)	

**Table 2 antioxidants-11-00138-t002:** Description of the whole sample across follow-up times (N:47). Statistically significant differences (*p*-value ≤ 0.05) are underlined in bold.

	T0		T3		*p*-Value
	Mean ± SD	Median (Min to Max)	Mean ± SD	Median (Min to Max)	
CDAI (Crohn’s Disease Activity Index score, CD patients)	288.09 (60.73)	286.5 (168 to 414)	209.68 (57.64)	204 (137 to 371)	** <0.0001 **
Mayo score (UC patients)	13.64 (2.78)	14 (7 to 19)	8.64 (2.75)	9 (4 to 14)	** <0.0001 **
GM-CSF (pg/mL)	1.29 ± 5.13	0 (0 to 32.37)	1 ± 4.84	0 (0 to 32.05)	0.14
IFN-α (pg/mL)	7.87 ± 4.65	9.05 (0 to 17.45)	7.38 ± 4.66	9.45 (0 to 14.65)	0.90
IFN-γ (pg/mL)	44.23 ± 120.39	0 (0 to 662.92)	41.85 ± 110.29	0 (0 to 662.92)	0.60
TNFα (pg/mL)	8.83 ± 36.19	0 (0 to 232.08)	6.54 ± 32.23	0 (0 to 217.51)	0.07
TNFα (detectable)	6 (12.50)		6 (12.50)		--
CRP (mg/dL)	1.59 ± 2.06	0.64 (0.05 to 7.85)	0.74 ± 1.19	0.15 (0.05 to 5.29)	** <0.01 **
IL-10 (pg/mL)	39.31 ± 73.16	30.56 (0 to 441.55)	41.24 ± 68.24	31.06 (0 to 431.89)	0.69
IL-12 (pg/mL)	53.54 ± 196.61	0 (0 to 1145.63)	44.03 ± 176.04	0 (0 to 1134.84)	0.42
IL-17A (pg/mL)	3.44 ± 5.92	2.3 (0 to 36.23)	3.35 ± 5.56	2.61 (0 to 34.69)	0.82
IL-2 (pg/mL)	0.48 ± 3.35	0 (0 to 23.19)	0.52 ± 3.57	0 (0 to 24.72)	0.95
IL-4 (pg/mL)	20.81 ± 37.44	5.18 (0 to 215.53)	18.52 ± 36.48	3.39 (0 to 228.54)	0.51
IL-5 (pg/mL)	9.84 ± 34.55	0 (0 to 199.9)	6.59 ± 23.85	0 (0 to 147.07)	0.10
IL-6 (pg/mL)	3.63 ± 18.02	0 (0 to 106.12)	1.01 ± 6.98	0 (0 to 48.38)	0.37
IL-9 (pg/mL)	15.72 ± 50.43	0 (0 to 299.73)	15.09 ± 47.5	0 (0 to 290.69)	0.75
IL-1β (pg/mL)	9.88 ± 21.29	1.78 (0 to 126.75)	23.28 ± 110.19	2.37 (0 to 765.91)	0.67
Folate (ng/mL)	6.87 ± 5.59	4.95 (1.9 to 28)	8.14 ± 6.63	6.4 (1.7 to 28)	** 0.05 **
Vitamin B12 (pg/mL)	267.75 ± 141.19	244.5 (69 to 682)	245.98 ± 114.12	220 (83 to 566)	** 0.03 **
Vitamin D (ng/mL)	21.59 ± 10.29	19.45 (4.8 to 65.5)	20.21 ± 8.03	20.2 (4.6 to 46.4)	0.12

**Table 3 antioxidants-11-00138-t003:** Description of the whole sample in T3 according to the type of treatment. (N: 47). Statistically significant differences (*p*-value ≤ 0.05) are underlined in bold.

	T3				
	RFT		RED		*p*-Value
	Mean ± SD	Median (Min to Max)	Mean ± SD	Median (Min to Max)	
Proportions (%)	27 (57.40)		20 (42.60)		
Age (years)	40.26 ± 12.33	38 (19 to 69)	45.4 ± 16.71	47.5 (17 to 70)	
Gender					
Male	15 (55.60)		8 (40.00)		
Female	12 (44.40)		12 (60.00)		0.29
Disease					
CD	13 (48.10)		9 (45.00)		0.83
UC	14 (51.90)		11 (55.00)		
GM-CSF (pg/mL)	1.78 ± 6.4	0 (0 to 32.05)	0 ± 0	0 (0 to 0)	** 0.02 **
IFN-α (pg/mL)	7.02 ± 4.69	9.45 (0 to 11.45)	7.59 ± 4.67	9.25 (0 to 14.65)	0.99
IFN-γ (pg/mL)	61.09 ± 140.83	0 (0 to 662.92)	15.71 ± 41.3	0 (0 to 153.83)	0.09
TNFα (Detectable)	6 (22.20)		--		** 0.03 **
CRP (mg/dL)	0.87 ± 1.07	0.31 (0.05 to 3.8)	0.56 ± 1.33	0.05 (0.05 to 5.29)	** 0.03 **
IL-10 (pg/mL)	55.45 ± 85.88	32.06 (0 to 431.89)	22.02 ± 27.41	0 (0 to 75.96)	0.12
IL-12 (pg/mL)	73.19 ± 231.55	0 (0 to 1134.84)	6.87 ± 23.05	0 (0 to 98.36)	0.32
IL-17A (pg/mL)	4.36 ± 6.91	2.76 (0 to 34.69)	1.88 ± 2.68	0 (0 to 10.4)	0.11
IL-2 (pg/mL)	0.92 ± 4.76	0 (0 to 24.72)	0 ± 0	0 (0 to 0)	0.41
IL-4 (pg/mL)	25.34 ± 45.52	10.55 (0 to 228.54)	7.4 ± 13.59	0 (0 to 49.54)	0.06
IL-5 (pg/mL)	11.1 ± 31.14	0 (0 to 147.07)	0.82 ± 3.68	0 (0 to 16.44)	0.09
IL-6 (pg/mL)	1.79 ± 9.31	0 (0 to 48.38)	0 ± 0	0 (0 to 0)	0.41
IL-9 (pg/mL)	23.13 ± 61.23	0 (0 to 290.69)	4.98 ± 15.46	0 (0 to 56.2)	0.19
IL-1β (pg/mL)	34.13 ± 146.54	1.93 (0 to 765.91)	6.34 ± 8.5	3.36 (0 to 25.93)	0.88
Folate (ng/mL)	7.07 ± 6.53	4.5 (1.7 to 28)	9.58 ± 6.65	7.05 (2.6 to 25)	0.06
Vitamin B12 (pg/mL)	260.44 ± 121.62	235 (83 to 533)	226.45 ± 102.92	207 (96 to 566)	0.32
Vitamin D (ng/mL)	20.07 ± 8.62	21.8 (4.6 to 39.3)	20.4 ± 7.38	19.45 (13.2 to 46.4)	0.69

**Table 4 antioxidants-11-00138-t004:** Description of the whole sample at follow-up observation (T3) in CD patients, according to the type of treatment. (N: 22). Statistically significant differences (*p*-value ≤ 0.05) are underlined in bold.

	CD				
	RFT		RED		
	Mean ± SD	Median	Mean ± SD	Median	*p*-Value
		(Min to Max)		(Min to Max)	
Proportions (%)	13 (59.10)		9 (40.90)		
Age (years)	37.8 ± 11.6	37 (19 to 59)	45.67 ± 18.25	47 (17 to 70)	0.20
Gender					
Male	7 (53.80)		5 (55.60)		0.95
Female	6 (46.20)		4 (44.40)		
GM-CSF (pg/mL)	2.89 ± 8.8	0 (0 to 32.05)	0 ± 0	0 (0 to 0)	** 0.04 **
IFN-α (pg/mL)	8.39 ± 3.79	9.85 (0 to 11.45)	7.66 ± 4.4	9.45 (0 to 11.45)	0.58
IFN-γ (pg/mL)	96.3 ± 185.69	0 (0 to 662.92)	0 ± 0	0 (0 to 0)	** 0.02 **
TNFα (pg/mL)	5 (38.50)		--		** 0.05 **
CRP (mg/dL)	0.98 ± 1.35	0.4 (0.05 to 3.8)	0.13 ± 0.12	0.05 (0.05 to 0.39)	** 0.03 **
IL-10 (pg/mL)	74.74 ± 113.94	31.06 (0 to 431.89)	8.79 ± 17.46	0 (0 to 41.06)	** 0.02 **
IL-12 (pg/mL)	119.04 ± 312.33	0 (0 to 1134.84)	0 ± 0	0 (0 to 0)	** 0.02 **
IL-17A (pg/mL)	7 ± 9.21	3.37 (0 to 34.69)	0.82 ± 1.65	0 (0 to 4.29)	** 0.01 **
IL-2 (pg/mL)	1.9 ± 6.86	0 (0 to 24.72)	0 ± 0	0 (0 to 0)	0.45
IL-4 (pg/mL)	37.69 ± 61.97	14.12 (0 to 228.54)	5.08 ± 6.38	0 (0 to 14.12)	0.10
IL-5 (pg/mL)	17.5 ± 40.14	0 (0 to 147.07)	0 ± 0	0 (0 to 0)	** 0.05 **
IL-6 (pg/mL)	3.72 ± 13.42	0 (0 to 48.38)	0 ± 0	0 (0 to 0)	0.45
IL-9 (pg/mL)	35.15 ± 79.49	0 (0 to 290.69)	0 ± 0	0 (0 to 0)	** 0.04 **
IL-1β (pg/mL)	7.87 ± 11.93	1.79 (0 to 34.62)	1.86 ± 2.62	0.18 (0 to 6.98)	0.32
Folate (ng/mL)	5.29 ± 2.01	6 (1.7 to 8.7)	12.23 ± 9.07	8.5 (2.7 to 25)	0.24
Vitamin B12 (pg/mL)	244.85 ± 113.11	235 (83 to 463)	216.89 ± 137.44	202 (96 to 566)	0.25
Vitamin D (ng/mL)	21.51 ± 9.96	23 (6.6 to 39.3)	21.4 ± 9.81	19.3 (13.5 to 46.4)	0.74

**Table 5 antioxidants-11-00138-t005:** Description of the whole sample of UC patients at the follow-up observation (T3) in UC patients, according to the type of treatment. (N: 25).

	UC				
	RFT		RED		
	Mean ± SD	Median	Mean ± SD	Median	*p*-Value
		(Min to Max)		(Min to Max)	
Proportions (%)	14 (56.00)		11 (44.00)		
Age (years)	42.57 ± 12.99	41 (25 to 69)	45.18 ± 16.25	48 (19 to 67)	0.62
Gender					
Male	6 (42.90)		4 (36.40)		0.30
Female	8 (57.10)		7 (63.60)		
IFN-α (pg/mL)	5.74 ± 5.2	8.85 (0 to 11.05)	7.53 ± 5.09	9.05 (0 to 14.65)	0.61
IFN-γ (pg/mL)	28.39 ± 73.81	1.6 (0 to 281.01)	28.57 ± 53.26	0 (0 to 153.83)	0.97
TNFα (pg/mL)	1 (7.10)		--		0.95
CRP (mg/dL)	0.77 ± 0.77	0.3 (0.05 to 1.9)	0.9 ± 1.75	0.05 (0.05 to 5.29)	0.28
IL-10 (pg/mL)	37.54 ± 45.27	32.06 (0 to 180.06)	32.84 ± 29.96	34.06 (0 to 75.96)	0.85
IL-12 (pg/mL)	30.62 ± 114.58	0 (0 to 428.72)	12.49 ± 30.54	0 (0 to 98.36)	0.24
IL-17A (pg/mL)	1.9 ± 1.94	1.8 (0 to 5.2)	2.76 ± 3.1	2.76 (0 to 10.4)	0.65
IL-2 (pg/mL)	0 ± 0	0 (0 to 0)	0 ± 0	0 (0 to 0)	--
IL-4 (pg/mL)	13.87 ± 17.44	6.97 (0 to 53.05)	9.29 ± 17.59	0 (0 to 49.54)	0.32
IL-5 (pg/mL)	5.17 ± 19.33	0 (0 to 72.33)	1.49 ± 4.96	0 (0 to 16.44)	0.95
IL-6 (pg/mL)	0 ± 0	0 (0 to 0)	0 ± 0	0 (0 to 0)	--
IL-9 (pg/mL)	11.97 ± 37.24	0 (0 to 138.54)	9.05 ± 20.34	0 (0 to 56.2)	0.83
IL-1β (pg/mL)	58.51 ± 203.68	2.37 (0 to 765.91)	10.01 ± 9.95	7.87 (0 to 25.93)	0.35
Folate (ng/mL)	8.73 ± 8.69	4.2 (2.8 to 28)	7.41 ± 2.58	7 (2.6 to 13)	0.41
Vitamin B12 (pg/mL)	274.93 ± 131.54	239.5 (114 to 533)	234.27 ± 69.76	230 (129 to 365)	0.72
Vitamin D (ng/mL)	18.73 ± 7.27	20.7 (4.6 to 28.1)	19.58 ± 5	20.2 (13.2 to 27.8)	0.97

## Data Availability

The data presented in this study are available on request from the corresponding author.

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
