# Peer review of "Nutritional Regimes Enriched with Antioxidants as an Efficient Adjuvant for IBD Patients under Infliximab Administration, a Pilot Study"

_antioxidants, 2022, doi:10.3390/antiox11010138_

Round 1

Reviewer 1 Report

The paper presents the result of a single-center preliminary study (SI-CURA) conducted to assess whether daily administration of purple a maize corn supplement could improve the response to infliximab in IBD patients with both Crohns’ disease and ulcerative colitis.

The study is well done, the data has been properly analyzed. This is a preliminary study, but it is suitable for publication in Antioxidants.

Introduction section: The authors substantiated the design of this interventional study by referring to the appropriate references.

Materials and methods: The study was well designed. IBD patients were correctly enrolled in the study. Dietary therapy management has been described. The methods were properly described.

The results are clearly presented and correctly analyzed. The discussion is factual, based on well-chosen literature. Conclusions are supported by the results.

Minor corrections:

Text formatting should be carefully checked.

The language should be modified carefully.

Author Response

We would like to thank REVIEWER 1 for his/her positive comments.

We hope that in the present form the manuscript will be considered suitable for publication

The paper presents the result of a single-center preliminary study (SI-CURA) conducted to assess whether daily administration of purple a maize corn supplement could improve the response to infliximab in IBD patients with both Crohns’ disease and ulcerative colitis.

The study is well done, the data has been properly analyzed. This is a preliminary study, but it is suitable for publication in Antioxidants.

Introduction section: The authors substantiated the design of this interventional study by referring to the appropriate references.

Materials and methods: The study was well designed. IBD patients were correctly enrolled in the study. Dietary therapy management has been described. The methods were properly described.

The results are clearly presented and correctly analyzed. The discussion is factual, based on well-chosen literature. Conclusions are supported by the results.

Minor corrections:

Text formatting should be carefully checked.

The language should be modified carefully.

We thank the Reviewer for his/her comments and appreciation of our work.

The text has been reformatted and edited by a native English speaker.

Reviewer 2 Report

The article by Liso, M. et al. studying antioxidant suplements for IBD patients under Infliximab treatment seems scientifically correct. The manuscript employs an analytical approach to study the efficacy of anthocyanin compounds to improve the course of IBD patients. Taken globally, I like the work, being an interesting focus the modulatation of the intestinal microbiota.

In addition I would like to state that considering the prevalence of IFD the number of patients is low: you indicate that this is a preliminary study which paves the way for a wider metacentric study, so I guess a more ambitious study is on its way. In this case, I strongly recommend to include endoscopy and an histological study comparing T0 and T3 in the next experiment. This point would decisively improve the design of the study. I´m sure that pathology may reveal very interesting findings when comparing both biopsies.

Regarding the use of English language I think it is good, with occassional minor faults. Probably the most frequent fault is a lack of commas. These points are addressed as minor faults.

In a purely scientific level, I think the manuscript has some major issues to be considered:

  • I miss a bit of text in the results section: there are big tables, with an excess of information that is undifferentiated in terms of relevance: some more text briefly explaining them would be appreciated. May be a new graph (or various if neccesary) can be added for a more obvious comparison between the most relevant variables, as long as Figure 2 is not highlighted from the background. You may consider to reduce all the tables retaining only the key information and take the less important information to the supplementary data section. May be including the comparison at the end of the results section would be fine. Probably you should think this point deeply to reach a greater impact.
  • In the discussion section, I would like to see a clearer discussion of the described findings (lines 304-3xx). What I miss in the current version is an explanation of the rationale behind the conclusions. You should include all the relevant analytical values and explain why these values are relevant. As I said previously, all the information is present in the graph and tables, but they give the same importance to the patient´s height and TNF values. Tables are exhaustive, and for this reason I consider the results are not clearly presented. Please improve this.
  • In the discussion (lines 278-279 and 332-340) you suggest a more potent effect in small intestine and CD. Even considering the preliminary nature of the present work, I would like to see a comparison between the clinical variants of UC (as long as all UC belong to the extensive pancolitis variant). Would it be possible that colic CD has a similar behaviour to UC?
  • Conclusions, lines 348-353. The current conclusions section seems like a personal impression of the authors. These impressions can be included (with modifications) in the discussion, but what is mandatory in this (optional) section is a final idea (or ideas) of the work. The substance that makes a scientific experiment be worthwhile.

I would like to propose various additional minor changes:

  • Abstract, lines 31-32. Begin the section with this affirmation undermines the interest of the whole article. You should begin with something more directly related with your work. In addition, inflammatory bowel diseases are actually characterized by various clinical and histological alterations. An increased incidence in western countries can be observed in many other pathological conditions. I really don´t like this sentence, this information is already present in the introduction. You should improve this section to make it more both “attractive” and “informative” about the manuscript.
  • Introduction, lines 64 and 67. Please include a comma here after “initial therapy”. In the next sentence please do the same after “IBD remisssion”.
  • Introduction, lines 72, 73 and 74. Please consider to include reference [8] at the end of the sentence. In line 73 please include a comma after “and indirectly”. Finally, regarding the sentence ending in line 74, I think that nutritional components can, directly or indirectly, favour or impair desired or indesired bacterial species. With this complicated sentence I mean that nutritional components may decisively influence the intestinal flora or microbiota.
  • Introduction, lines 84-85. I think it would be more elegant to type “reduce iron concentration by inducing the expression of […] and […]” to avoid the reiteration of the word “and”.
  • Introduction, line 100-101. I think “damage” would be best suited to the sentence than “damages”. I also think that this sentence is too complex can be divided. I suggest a division after citation [31].
  • Introduction, lines 105-111. The whole paragraph has no place in the introduction section. You may consider to include it as a summary after the discussion or may be as a separate conclusion.
  • Materials and methods, line 168. I think that the bromatological composition of the diet may be included as supplementary material. It is purely descriptive and has very little relationship with the results.
  • Materials and methods, line 175-176. Patients are assumed to be blinded. You should specify if their physicians were blinded and if the final analysts were blinded too.
  • Materials and methods, line 185. What a certain reagent is does not form part of the materials and methods section. In any case you should include all the information about the purple corn supplement in the introduction section: you may add new information if it´s your will.
  • Materials and methods, line 190. Here I think there is a little fault, you tiped “die” but probably you mean “day”.

  • Results, line 228. I think you can remove “enrolled” from this second sentence.
  • Results, Table 3. At the end of the table there is an asterisk, but I can see no asterisk in the table referred to the previous asterisk. If the Wilcoxon test is applied to the whole table, I think this point should be better explained in the text. The same for Tables 4, 5 and 6.
  • Results, lines 264-265. This first sentence is reiterative. You can remove it.
  • Discussion, lines 276-280. This paragraph is a conclusion. You should not begin the discussion section with it.
  • Discussion, lines 304-310. Here you are in the discussion section, you don´t need to recapitulate the design of your study
  • Discussion, lines 310-322. A great proportion of this text seems more correct in the results section. You can fill the lack of text in that section with this (and you can even explain more findings of your experiment).
  • Discussion, lines 332-334. You should specify the results you are talking about.
  • Results, lines 346-349. Please consider the division of this sentence. It is too complicated to read.
  • Results, lines 358-360. Please, improve the sentence.
  • Discussion, line 423. This is the second “axons” in the sencence. Could it be “cytoplasms”?
  • Discussion, lines 452-461. I feel this paragraph should be either in “supplementary materials” or in “data availability statement”.

Author Response

REVIEWER 2

The article by Liso, M. et al. studying antioxidant suplements for IBD patients under Infliximab treatment seems scientifically correct. The manuscript employs an analytical approach to study the efficacy of anthocyanin compounds to improve the course of IBD patients. Taken globally, I like the work, being an interesting focus the modulatation of the intestinal microbiota.

First of all, we are grateful to the reviewer for the constructive work done with the intent to improve our manuscript. We have changed our manuscript according to the reviewers’ suggestions.

In addition I would like to state that considering the prevalence of IFD the number of patients is low: you indicate that this is a preliminary study which paves the way for a wider metacentric study, so I guess a more ambitious study is on its way. In this case, I strongly recommend to include endoscopy and an histological study comparing T0 and T3 in the next experiment. This point would decisively improve the design of the study. I´m sure that pathology may reveal very interesting findings when comparing both biopsies.

We thank the Reviewer for his/her comments and suggestions for the next experimental phase of our study. We will take into consideration the possibility of histological analysis at T0 and T3 for the next experiments.

Regarding the use of English language I think it is good, with occassional minor faults. Probably the most frequent fault is a lack of commas. These points are addressed as minor faults.

The text has been reformatted and edited by a native English speaker.

In a purely scientific level, I think the manuscript has some major issues to be considered:

  • I miss a bit of text in the results section: there are big tables, with an excess of information that is undifferentiated in terms of relevance: some more text briefly explaining them would be appreciated. May be a new graph (or various if neccesary) can be added for a more obvious comparison between the most relevant variables, as long as Figure 2 is not highlighted from the background. You may consider to reduce all the tables retaining only the key information and take the less important information to the supplementary data section. May be including the comparison at the end of the results section would be fine. Probably you should think this point deeply to reach a greater impact.
  • We agree with the Reviewer and have subdivided Tables 3-4-5-6 (New Tables S2-S3-S4-S5), separating the physical and hematological data from circulating cytokines. New Supplementary Tables (S2-S3-S4-S5) have been added.
  • In the discussion section, I would like to see a clearer discussion of the described findings (lines 304-3xx). What I miss in the current version is an explanation of the rationale behind the conclusions. You should include all the relevant analytical values and explain why these values are relevant. As I said previously, all the information is present in the graph and tables, but they give the same importance to the patient´s height and TNF values. Tables are exhaustive, and for this reason I consider the results are not clearly presented. Please improve this.
  • We have improved the Discussion section and reformatted all the Tables.
  • In the discussion (lines 278-279 and 332-340) you suggest a more potent effect in small intestine and CD. Even considering the preliminary nature of the present work, I would like to see a comparison between the clinical variants of UC (as long as all UC belong to the extensive pancolitis variant). Would it be possible that colic CD has a similar behaviour to UC?
  • We have added this information in the Discussion, and added the related references (53-54).
  • Conclusions, lines 348-353. The current conclusions section seems like a personal impression of the authors. These impressions can be included (with modifications) in the discussion, but what is mandatory in this (optional) section is a final idea (or ideas) of the work. The substance that makes a scientific experiment be worthwhile.
  • We have improved the Conclusions section, adding a summary of the main findings.

I would like to propose various additional minor changes:

  • Abstract, lines 31-32. Begin the section with this affirmation undermines the interest of the whole article. You should begin with something more directly related with your work. In addition, inflammatory bowel diseases are actually characterized by various clinical and histological alterations. An increased incidence in western countries can be observed in many other pathological conditions. I really don´t like this sentence, this information is already present in the introduction. You should improve this section to make it more both “attractive” and “informative” about the manuscript.
  • We have improved the Abstract, particularly in the “Background” section.
  • Introduction, lines 64 and 67. Please include a comma here after “initial therapy”. In the next sentence please do the same after “IBD remisssion”.
  • We have added a comma in lines 64 and 67.
  • Introduction, lines 72, 73 and 74. Please consider to include reference [8] at the end of the sentence. In line 73 please include a comma after “and indirectly”. Finally, regarding the sentence ending in line 74, I think that nutritional components can, directly or indirectly, favour or impair desired or indesired bacterial species. With this complicated sentence I mean that nutritional components may decisively influence the intestinal flora or microbiota.
  • We have changed the text according to the Reviewer's suggestions.
  • Introduction, lines 84-85. I think it would be more elegant to type “reduce iron concentration by inducing the expression of […] and […]” to avoid the reiteration of the word “and”.
  • We have corrected the text.
  • Introduction, line 100-101. I think “damage” would be best suited to the sentence than “damages”. I also think that this sentence is too complex can be divided. I suggest a division after citation [31].
  • We have changed the text in lines 100-101.
  • Introduction, lines 105-111. The whole paragraph has no place in the introduction section. You may consider to include it as a summary after the discussion or may be as a separate conclusion.
  • We have moved the text in lines 105-111 to the Conclusions paragraph.
  • Materials and methods, line 168. I think that the bromatological composition of the diet may be included as supplementary material. It is purely descriptive and has very little relationship with the results.
  • We have moved Table 1 to the Supplementary file (New Table S1).
  • Materials and methods, line 175-176. Patients are assumed to be blinded. You should specify if their physicians were blinded and if the final analysts were blinded too.
  • We have added this information in paragraph 2.4.
  • Materials and methods, line 185. What a certain reagent is does not form part of the materials and methods section. In any case you should include all the information about the purple corn supplement in the introduction section: you may add new information if it´s your will.
  • We have moved line 185 from Material and Methods to the Introduction section.
  • Materials and methods, line 190. Here I think there is a little fault, you tiped “die” but probably you mean “day”.
  • We have changed the Latin word “die” to “day”. 
  • Results, line 228. I think you can remove “enrolled” from this second sentence.
  • We have made this correction.
  • Results, Table 3. At the end of the table there is an asterisk, but I can see no asterisk in the table referred to the previous asterisk. If the Wilcoxon test is applied to the whole table, I think this point should be better explained in the text. The same for Tables 4, 5 and 6.
  • We have removed the sentence with an asterisk from the end of the Tables.
  • Results, lines 264-265. This first sentence is reiterative. You can remove it.
  • We have removed the phrase according to the Reviewer's suggestion.
  • Discussion, lines 276-280. This paragraph is a conclusion. You should not begin the discussion section with it.
  • We have removed lines 276-280 from the Discussion section.
  • Discussion, lines 304-310. Here you are in the discussion section, you don´t need to recapitulate the design of your study
  • We have removed lines 304-307 from the Discussion section.
  • Discussion, lines 310-322. A great proportion of this text seems more correct in the results section. You can fill the lack of text in that section with this (and you can even explain more findings of your experiment).
  • We have moved the text in lines 310-322 to the Results section (paragraph 3.2).
  • Discussion, lines 332-334. You should specify the results you are talking about.
  • We have rephrased lines 332-334, specifying the discussed results.
  • Results, lines 346-349. Please consider the division of this sentence. It is too complicated to read.
  • Results, lines 358-360. Please, improve the sentence.
  • Discussion, line 423. This is the second “axons” in the sencence. Could it be “cytoplasms”?
  • Discussion, lines 452-461. I feel this paragraph should be either in “supplementary materials” or in “data availability statement”.
  • We think that these comments were included by mistake by the reviewer as these lines do not appear in the present work.
